# The Interaction between microRNAs and the Wnt/β-Catenin Signaling Pathway in Osteoarthritis

**DOI:** 10.3390/ijms22189887

**Published:** 2021-09-13

**Authors:** Xiaobin Shang, Kai Oliver Böker, Shahed Taheri, Thelonius Hawellek, Wolfgang Lehmann, Arndt F. Schilling

**Affiliations:** Department of Trauma Surgery, Orthopaedics and Plastic Surgery, University Medical Center Goettingen, Robert Koch Straße 40, 37075 Göttingen, Germany; xiaobin.shang@stud.uni-goettingen.de (X.S.); shahed.taheri@med.uni-goettingen.de (S.T.); Thelonius.hawellek@med.uni-goettingen.de (T.H.); Wolfgang.Lehmann@med.uni-goettingen.de (W.L.); Arndt.schilling@med.uni-goettingen.de (A.F.S.)

**Keywords:** osteoarthritis, Wnt/β-catenin, miRNA, non-coding RNA

## Abstract

Osteoarthritis (OA) is a chronic disease affecting the whole joint, which still lacks a disease-modifying treatment. This suggests an incomplete understanding of underlying molecular mechanisms. The Wnt/β-catenin pathway is involved in different pathophysiological processes of OA. Interestingly, both excessive stimulation and suppression of this pathway can contribute to the pathogenesis of OA. microRNAs have been shown to regulate different cellular processes in different diseases, including the metabolic activity of chondrocytes and osteocytes. To bridge these findings, here we attempt to give a conclusive overview of microRNA regulation of the Wnt/β-catenin pathway in bone and cartilage, which may provide insights to advance the development of miRNA-based therapeutics for OA treatment.

## 1. Introduction

Osteoarthritis (OA) is one of the most common diseases affecting millions of people worldwide. The typical symptoms are pain, swelling, stiffness, and loss of flexibility in weight-bearing joints, especially knee and hip joints, breakdown of cartilage, and finally, the loss of joint function, which seriously affects patients’ life and work. Around 14 million cases in the US alone suffer from symptomatic OA, and the cost of OA treatment is a heavy social burden [1]. The risk factors can be classified into two categories: person-level factors such as age, gender, obesity, and genetics, and joint-level factors such as injury, malalignment, and abnormal mechanical stress. Conventional treatment methods include physical therapy based on rehabilitation exercises and medication, especially non-steroidal anti-inflammatory drugs, all of which can only relieve the symptoms but cannot reverse the course of the disease. For the end-stage patients, joint replacement surgery is the only effective treatment so far, although it comes with some disadvantages, including high cost, risk of perioperative complications and postoperative periprosthetic infection, the potential necessity of arthroplasty revision, and other problems [2]. The fundamental reason for limited treatment strategies for a disease is an insufficient understanding of its underlying pathogenesis. Recent studies have demonstrated that the Wnt/β-catenin pathway is involved in the pathophysiological processes of OA and that both the excessive stimulation and the suppression of this pathway can contribute to OA development [3]. In this review, we summarize the current understanding of how miRNAs influence Wnt/β-catenin signaling in OA development and the ensuing consequences for its pathogenesis and possible treatment.

## 2. miRNA Biogenesis and Target Prediction 

microRNAs (miRNAs) are a kind of small non-coding RNAs that are responsible for post-transcriptional regulation of 60% of human genes [4]. Since they were first discovered in 1993, miRNAs have been shown to participate in the modulation of diverse biological processes. More importantly, the pathologic expression of miRNAs can lead to various diseases, including OA [5]. The biogenesis of miRNAs has been deeply researched in the past decades. In brief, miRNAs are first transcribed from their genes into primary miRNAs (pri-miRNAs) with the support of RNA polymerase II (Pol II). Thereafter, pri-miRNAs are modified into precursor miRNAs (pre-miRNAs) by the microprocessor complex consisting of DiGeorge Syndrome Critical Region Gene 8 (DGCR8) and Drosha in the nucleus, followed by the transportation to the cytoplasm by Exportin 5. The cytoplasmic pre-miRNAs are then further processed into mature miRNAs duplex (20–23bp) by the RNase III enzyme Dicer. In most cases, one strand of the miRNAs duplex is incorporated into the RNA-induced silencing complex (RISC), where the miRNAs interact with the 3′ UTR of target mRNAs and thus suppress their expression. As a result, gene silencing happens through mRNA degradation or suppressed translation of the mRNA, depending on the ratio of complementarity between the miRNA and the target mRNA sequence. It was reported that the human genome might encode more than 1900 miRNAs to modify gene expression [6]. Public online websites such as “Targetscan”, “miRWalk”, and “miRDB” provide a prediction of possible targets of miRNAs through the interaction with the 3′ UTR of target mRNAs. However, prediction of the effect of such miRNAs silencing of their mRNA targets is complicated, considering each miRNA has multiple mRNA targets and vice versa. This can lead to complex effects already inside the single cell. It becomes even more complicated on a tissue level, where the same miRNA may have different effects in the different cells of the tissue.

## 3. miRNAs Regulation in Osteoarthritis

miRNAs mediated regulation of gene expression at the post-transcriptional level may play an important part in the transition from healthy joint to OA (Figure 1).

Indeed, during the past decades, abnormal expression of miRNAs has been demonstrated in OA patients compared to healthy cohorts [7]. Furthermore, miRNAs have been demonstrated to be able to play either a protective or destructive role in the pathogenesis of OA [8]. For example, it was reported that the expression of miR-140 was reduced in cartilage from OA patients compared to healthy cartilage, while intra-articular administration of miRNA-140 significantly relieved OA progression by maintaining cartilage homeostasis [9]. Similarly, miR-30a expression was decreased in OA chondrocytes, and ADAMTS-5 was identified as its direct target [10]. Additionally, MMP-19 was identified as the direct target of miR-193b-3p, and the miR-193b-3p expression decreased in OA chondrocytes, while treatment with an exogenous supplement of miR-193b-3p could relieve IL-1β-induced inflammatory reactions [11]. Although the above-mentioned miRNAs demonstrate a protective effect on cartilage, other miRNAs can shift the homeostasis of chondrocytes and cartilage towards a catabolic phenotype. For example, miR-365 expression was shown to be regulated by mechanical stress and upregulated cartilage extracted from primary and traumatic OA patients. A catabolic character of miR-365 could be demonstrated to be mediated via the inhibition of histone deacetylase 4 (HDAC4) [12]. Indeed, inhibition of miR-365 significantly decreased the MMP-13 and Col-X gene expression in IL-1β induced chondrocytes [13]. The results suggested that miR-365 inhibitors can be promising therapeutic targets for the treatment of OA. However, it seems unreasonable to treat OA only through the regulation of a single miRNA and then attempt to explain the combined function of one miRNA by one of the various targets. As there are multiple targets for the same miRNA, the same miRNA may also have different effects on OA progression by inhibiting various targets. For example, the function of miR-146a in arthritis pathology is controversial, as it was upregulated in OA cartilage and contributed to cartilage degeneration while downregulated in end-stage OA [14]. A possible explanation is that miR-146a targets different mRNAs at different stages of OA. Similarly, one mRNA can also be targeted by different miRNAs. For example, it has been shown that HDAC-4 can be targeted by at least miR-222, miR-140, and miR-365 at the same time [15,16,17].

## 4. Wnt Pathway Overview

Since its first discovery in 1982, the Wnt signaling pathway has been deeply studied in different diseases, especially in oncology [18]. Up to now, more than 19 Wnt proteins, as well as various Wnt receptors including Frizzled (FZD) receptors and low-density lipoprotein receptor-related proteins 5/6 (LRP5/6) coreceptors are shown to be involved in the singling pathway [19]. Based on the type of Wnt–FZD interactions, as well as the downstream involvement or absence of β-catenin, the Wnt pathway can be classified into two distinct downstream branches: the β-catenin-dependent canonical pathway and two β-catenin independent noncanonical pathways. In the canonical β-catenin-dependent Wnt pathway, β-catenin would be degraded through β-transducin repeat-containing protein (β-TrCP)-mediated ubiquitination in the cytoplasm, in case Wnt is absent or the normal Wnt–FZD interaction is blocked by antagonists. This process is activated by the enzyme glycogen synthase kinase 3β (GSK3β) in a “destruction complex” comprised of scaffold protein Axin, adenomatosis polyposis coli (APC), and kinases such as GSK-3β and casein kinase 1α (CK1α) [20,21]. Recent studies have found that yes-associated protein (YAP) and Tafazzin (TAZ), which are transcriptional regulators of the Hippo pathway, might also participate in the regulation of Wnt signaling by being an integral part of the β-catenin destruction complex [22]. Ligands such as Wnt3A are responsible for activating the canonical Wnt pathway [23]. When the Wnt pathway is activated, the Wnt−FDZ combination inactivates the destruction complex by promoting CK1α- and GSK-3β–mediated LRP phosphorylation, which leads to the polymerization and activation of Disheveled (Dsh). As a result, β-catenin is protected from degradation and is aggregated in the nucleus to stimulate transcription of downstream genes by activating the formation of β catenin–TCF/LEF (T-cell factor/lymphoid enhancing factor) complex [24]. The noncanonical Wnt signal transduction can be further classified into Wnt/Ca2+ and planar cell polarity (PCP) pathways based on different downstream signals, both of which could participate in the regulation of cytoskeletal organization, cell differentiation, and communication [25]. Wnt signaling can also be regulated by various antagonists such as Dickkopf (DKK), Wnt inhibitory factor 1 (WIF1), sclerostin, secreted FZD-related proteins (SFRPs), and insulin-like growth factor-binding protein 4 (IGFBP4) [26]. These antagonists can block Wnt signaling pathways, but sometimes conflicting effects occur due to complex intrinsic mechanisms [27]. Additionally, these so-called antagonists can even act as agonists depending on the cellular context [28].

## 5. Wnt Pathway Regulation in Osteoarthritis (OA)

The Wnt pathway regulates complex cellular processes such as proliferation, differentiation, regeneration, aging, and apoptosis during the embryologic development and pathogenesis of diseases. Variants in Wnt pathway-related genes have been indicated as susceptibility factors for OA [29]. The active Wnt pathway is crucial in chondrocyte hypertrophy: an important pathological process in OA cartilage [20]. Canonical Wnt signaling can contribute to OA, as both excessive activation and inactivation of β-catenin in cartilage seem to contribute to osteoarthritic changes in animal experiments. Zhu et al. reported that activation of β-catenin in articular cartilage chondrocytes induced premature chondrocyte differentiation into an OA-like phenotype in adult mice [30]. On the other hand, the team reported that the inhibition of β-catenin signaling in chondrocytes promoted cell apoptosis, which also led to articular cartilage destruction [31]. Similarly, Lietman et al. found that inhibition of the Wnt pathway could relieve OA symptoms in a surgery-induced murine OA model [32], while another study showed opposite results, namely, the transient activation of the Wnt pathway could induce or even accelerate repair of the damaged articular cartilage [33]. Accordingly, it is reasonable to deduce that moderate Wnt activity is necessary for the physiological maintenance of cartilage. A recent study found a buffering function for Wnt16, which may explain how a fine balance of Wnt activation is achieved in highly dynamic biological processes such as cartilage damage and/or repair in OA [34].

It is also known that the Wnt pathway can affect cartilage indirectly by regulating the structure of subchondral bone [35]. For example, it was found that selective inhibition by Dkk-1 in bone could decrease OA severity through VEGF inhibition, which in turn decreased the expression of matrix metalloproteinases in chondrocytes [27]. This study showed that the Wnt pathway not only participated in the pathophysiology of cartilage or subchondral bone but also played a complex role for the whole joint. Therefore, the question of how to accurately regulate Wnt signaling may even be more complicated than initially thought. Nevertheless, some inhibitors of the Wnt pathway have already been tested in early clinical trials, showing promising results. For example, treatment with XAV-939 or SM04690 was shown to ameliorate OA severity associated with reduced cartilage degeneration in vivo [32,36]. SM04690 even demonstrated remarkable pain relief and improved function in a completed clinical phase I trial [37].

## 6. miRNAs Modulate the Wnt Pathway in OA

miRNAs can activate or repress components of the Wnt pathway and thereby act on different targets in OA. Conversely, the Wnt pathway regulates the expression of miRNAs. For OA, this is especially important in processes maintaining joint homeostases, such as cartilage synthesis and degradation, inflammation, chondrocyte proliferation, differentiation, and apoptosis. The intracellular regulatory mechanisms of miRNAs on canonical Wnt pathways in OA are illustrated in Figure 2. miRNAs are associated with various components of Wnt signaling during OA progression, summarized in the following chapters.

## 7. Targeting Wnt Ligands/Receptors and Associated Proteins

Both Wnt-ligands and Wnt-receptors can be modulated by miRNAs. Table 1 summarizes miRNAs that target Wnt ligand/receptors, as well as their associated proteins in the pathogenesis of OA.

MiR-497-5p expression was found to be downregulated in OA cartilage and IL-1β-induced chondrocytes. Wnt3a was shown to be a direct target of miR-497-5p [38]. Indeed, the overexpression of miR-497-5p prominently upregulated the expression of cartilage matrix ingredients such as collagen II and aggrecan, reduced the expression of catabolic proteinases including MMP13 and ADAMTS4, while Wnt3a overexpression could reverse these effects. It was concluded that miR-497-5p attenuated the IL-1β-induced cartilage matrix degradation in chondrocytes through Wnt pathway regulation [38]. Similarly, miR-410 could regulate chondrogenic differentiation of mesenchymal stem cells (MSCs) by directly targeting Wnt3a, whose expression was upregulated in the cartilage of OA patients but showed a significant negative correlation with miR-410 expression [39]. Recently, many studies have reported that MSC-derived exosomes might participate in regulating biological processes such as cell migration, proliferation, and differentiation. For example, miR-92a-3p expression was demonstrated to be significantly reduced in OA chondrocyte-secreted exosomes compared with healthy controls, while treatment with MSC-miR-92a-3p-Exos was shown to promote cell proliferation and expression of matrix genes such as aggrecan, COL2A1, COL9A1, COMP, and SOX9 in MSCs [40]. Intriguingly, the regulation of cartilage homeostasis could be explained by exosomal miR-92a-3p mediated modulation of the Wnt pathway by directly targeting Wnt5a [40]. Similarly, miR-26b and miR-154-5p also participated in the pathophysiological process of OA by modulating the Wnt pathway [41,42].

Mechanical loading is important for the physiological and pathological metabolism of both cartilage and subchondral bone. Recent studies have shown that mechanosensitive miRNAs play a crucial role in the regulation of osteogenic/chondrogenic differentiation through mechanical cues and thus are important for bone/cartilage homeostasis [49]. For example, it was reported that miR-154-5p could inhibit osteogenic differentiation from adipose-derived mesenchymal stem cells (ADSCs) through the Wnt/PCP pathway by directly targeting Wnt11 under tensile stress [42]. This could provide an interconnection between the molecular mechanism of mechanosensitive miRNAs and the Wnt pathway in the osteogenic differentiation process of ADSCs. Additionally, other miRNAs can indirectly regulate Wnt by targeting related mRNAs. For example, Myeloid cell leukemia 1 (Mcl-1) was identified as a target of miR-203 [43]. It was demonstrated that knee joint immobility could induce Mcl-1 overexpression in articular cartilage chondrocytes [50]. It was also found that overexpression of Mcl-1 relieved the inactivation of Wnt/β-catenin pathways in lipopolysaccharides (LPS)-treated chondrocytes, while inactivation of Mcl-1 showed opposite effects [43]. Similarly, miR-146a could indirectly regulate the inflammatory reactions of chondrocytes via targeting CXCR4, which was related to the expression of Wnt/β-catenin signal factors, even though the exact function of miR-146a in arthritis pathology is still disputed [44].

miRNAs that regulate Wnt receptors also contribute to the development and progression of OA. It was shown that expression of the miR-29 family was downregulated in OA cartilage, and FZD3, FZD5, DVL3, FRAT2, and CK2A2 were validated as direct targets of the miR-29 family [45]. All of the targets belonged to positive regulators of the Wnt pathway, and targeting them seemed to modulate the Wnt signaling negatively. In another line of research, miR-1 was demonstrated to inhibit chondrocyte degradation through FZD7 [46].

Aside from the direct inhibition of Wnt pathway components, miRNAs that regulate its inhibitory factors may also be able to modulate this pathway. In this respect, Dickkopf-1 (DKK1) has been extensively studied [51]. It was reported that miR-335-5p could downregulate DKK1 and therefore activate Wnt signaling, a process that is likely to affect subchondral bone development [47]. As an activator of the Wnt pathway, Polycomb group protein BMI1 inhibits the DKK family and is overexpressed in OA chondrocytes [52]. miR-320a was reported to regulate the expression of BMI-1 in chondrocytes and thereby protect against cartilage degeneration [48].

## 8. Targeting β-catenin Destruction Complex and Associated Proteins

As one of the most important components in the Wnt pathway, β-catenin functions by accumulation in the cytoplasm and then translocation into the nucleus. Afterwards, it directly binds to TCF/LEF transcription factors while turning on downstream gene transcription. Finally, β-catenin transduces extracellular signals and modulates the response of receiving cells. In the context of OA, it can contribute to its pathogenesis by regulating different cellular processes such as cell proliferation, cell migration, and tissue regeneration. It has been reported that several miRNAs that are involved in osteoarthritis could inhibit the Wnt pathway by targeting β-catenin directly (Table 2).

MiR-320c has been well-studied in recent years due to its significant regulatory functions in different fields such as oncology and the musculoskeletal system [53]. Previous studies demonstrated that miR-320 could protect chondrocytes from degradation by suppressing β-catenin in the cytoplasm. Furthermore, miR-320 inhibited the transcriptional activity of the β-catenin/TCF complex in the nucleus, while intra-articular injection of miR-320-3p in a collagenase-induced OA model significantly reduced the expression of β-catenin, relieved OA progression, and re-established the smooth surface of cartilage [60]. In another study, miR-10a was able to suppress the Wnt/β-catenin pathway by decreasing β-catenin expression, while an exact mechanism could not be proposed [54]. Similarly, miR-200a-3p could regulate β-catenin expression indirectly by targeting Forkhead box C1 (FoxC1) in OA synovial fibroblasts and thus alleviate inflammation of the synovial membrane in OA joints [55].

When Wnt proteins are absent, β-catenin is normally degraded by the “destruction complex”; therefore, the miRNAs targeting APC, GSKβ, AXIN, and associated factors (see Table 2 for details) can modulate the Wnt pathway as described below. APC is a scaffolding protein for the destruction complex and has been identified to be a target of miR-142-3p. Specifically, miR-142-3p could stimulate the Wnt pathway via the inhibition of APC, which eventually resulted in the aggregation and nuclear translocation of β-catenin. Consequently, miR-142-3p could be a vital mediator of osteoblast differentiation and even be exploited in a new therapeutic strategy to treat subchondral bone sclerosis in end-stage OA [56]. As for the kinase in the “destruction complex”, GSK-3β was predicted and validated as a target of miR-26ab through biological experiments. Studies have suggested that miR-26b could enhance the osteogenic potential of bone marrow-derived mesenchymal stem cells (BMSCs) by directly targeting GSK3β, and thus, activating Wnt signaling [57]. The Wnt pathway was also demonstrated to regulate the transcription of yes-associated protein (YAP) in colorectal carcinoma cells, while nuclear accumulation of YAP was correlated with β-catenin activation [61]. This is consistent with previous experiments that concluded YAP might modulate the Wnt pathway by being an integral part of the “destruction complex” [62]. In another line of research, miR-195-5p directly inhibited REGγ (also known as PA28γ, PAME3, Ki antigen), which could positively regulate YAP, whereas miR-195-5p inhibitor was demonstrated to protect chondrocytes from apoptosis and inflammatory responses via modulating the Wnt- and NF-κB pathways [58]. In canonical Wnt signaling, Disheveled is found to be recruited by the Wnt receptor Frizzled while inactivating the destruction complex thereafter [63]. Furthermore, Disheveled was identified as a target of miR-29c-3p, which could further inhibit the osteogenic differentiation of BMSCs [59].

## 9. Targeting Wnt Pathway Transcription Factors and Associated Proteins

Wnt signaling acts through the binding of β-catenin to the transcriptional complex TCF/LEF. As a result, miRNAs that negatively regulate the mRNA level of TCF/LEF, and its associated co-activators/co-repressors may also modulate the signal transduction of the Wnt pathway. Mammals have a total of four TCF/LEF family members: TCF1, TCF3, TCF4, and LEF-1. It was shown that LEF-1 could be targeted by miR-449a, which further repressed the transcription of Sox 9, subsequently affecting the differentiation and chondrogenesis of human MSCs [64]. In another study, microRNA-449a was found to have an increased expression in OA chondrocytes, while microRNA-449a inhibition had a protective effect by targeting SIRT1 instead of LEF-1 [65]. The different targets suggest that cell types and specific conditions can influence the exact functions of microRNA-449a. Similarly, overexpression of miR-29a was found in chondrocytes, which significantly stimulated the activity of the Wnt pathway and promoted cell apoptosis by repression of Foxo3a. This process was shown to compete with TCF4 for interaction with β-catenin [66]. In addition to the TCF/LEF family, other transcriptional factors and co-activators/co-repressors such as HDAC, NLK, and SOX-9 can also be targeted by miRNAs to affect the Wnt pathway (Table 3). 

Recent studies have suggested that histone deacetylases (HDACs) were upregulated in inflammatory chondrocytes, while miRNAs that inhibited the HDACs expression could act as therapeutic targets for OA. It was reported that HDAC participates in the regulation of the Wnt pathway by protecting TCF4 from degradation, while aberrant HDAC expression has been reported in several diseases, including OA [75]. Moreover, the intraarticular injection of trichostatin A—an HDAC inhibitor that promotes the degradation of TCF4—was shown to protect cartilage from destruction in a surgically-induced OA model [76]. Similar to TSA, miR-222 could dramatically suppress HDAC-4 expression in IL-1β-treated chondrocytes, which indicated that miR-222 might function as a potential HDAC-4 inhibitor for the treatment of OA [17]. Additionally, miR-140, miR-365, miR-92a-3p, and miR-193b-3p were demonstrated to modulate Wnt signaling by targeting the 3′UTR of HDAC in OA chondrocytes [15,16,67,68].

NIMA-related kinase2 (NEK2) is a serine/threonine kinase that can regulate the cell cycle and associated gene expressions through positive regulation of the Wnt pathway by phosphorylating LEF1 and, therefore, can activate downstream transcription [77]. One study has reported that in OA chondrocytes, miR-138 expression was downregulated, NEK2 expression was upregulated, while Wnt signaling was activated. Ultimately, it was confirmed that miR-138 mimic could reverse the phenomenon by direct targeting NEK2 [69]. 

As a member of the SRY-box (Sox) containing gene family, Sox9 is essential for chondrocyte differentiation and chondrogenesis. It was demonstrated that the activated Wnt pathway could modulate the expression of SOX9 [78], while abnormal expression of SOX9 could, in turn, affect the Wnt pathway by promoting β-catenin phosphorylation and degradation in the nucleus [79]. MiR-101, miR-145, miR-615-3p, and miR-140-5p were shown to inhibit extracellular matrix production in chondrocytes by inhibition of SOX9, during which miR-101 and miR-145 targeted SOX9 directly [70,71,72,73,74].

## 10. miRNAs Targeted by Canonical Wnt Pathway in OA 

miRNA expression can be highly dynamic, tissue-specific, and dependent on external stimuli and the developmental stage. While there are sample data on miRNAs regulating the Wnt pathway, the reverse scenario (i.e., regulation of miRNA by components of the Wnt-pathway) is less studied. In Table 4, we collected reports on such miRNAs that are regulated by molecules of the canonical Wnt pathway in OA. 

It was reported that hydrostatic pressure (HP) was able to affect the expression of several miRNAs such as miR-27a/b, miR-140, miR-146a/b, and miR-365 and their associated targets including MMP-13, ADAMTS-5, and HDAC-4 [80]. This modulation could be explained by HP-induced inhibition of the Wnt pathway, the reduction of β-catenin inside the nuclei, and the increased accumulation in the cytoplasm under HP exposure [80]. miR-222 could inhibit the Wnt pathway by targeting HDAC and further promoting TCF4 degradation. Intriguingly, it was reported that treatment of OA fibroblast-like synoviocytes (OA-FLS) with HDAC inhibitors SAHA and LBH589 could increase miR-146a expression [17,81]. The anti-inflammatory effects of HDAC inhibitors can therefore be explained via crosstalk between miRNAs and the Wnt pathway and further manifest their potential application for OA treatment. Furthermore, accumulating evidence suggests that activation of the Wnt pathway could increase the expression of miR-29a, while miR-29a could, in turn, promote Wnt/β-catenin signaling as discussed before [66,82,84]. This means that miR-29a and Wnt pathways might produce a positive regulatory feedback loop in OA. In addition, it was found that miR-140 was uniquely expressed in chondrocytes and suppressed by the Wnt pathway-mediated inhibition of Sox9 [83]. 

## 11. miRNA-Based Therapeutics in OA

The typical current pharmaceutical treatment of OA consists of unspecific inhibition of inflammation, mainly through non-steroidal anti-inflammatory drugs (NSAIDs) that target a single inflammatory mediator: Cox-2. The regulation of upstream targets might offer more attractive therapeutic possibilities. Furthermore, as the typical pathological changes seen in OA-stricken joints are the destruction of articular cartilage and the thickening of the subchondral bone, the related miRNAs that regulate the pathogenesis could be potential therapeutic compounds. For example, it was demonstrated that the injection of lentiviruses containing miR-222 into mouse knee joints after destabilization of the medial meniscus could significantly reduce cartilage destruction [17]. Aside from the direct injection treatment, increasing evidence has shown that extracellular vesicles may participate in progenitor cell-mediated tissue regeneration as a paracrine-driven mechanism. For example, overexpressed miR-140-5p in exosomes secreted by synovial mesenchymal stem cells (SMSCs) was able to promote the proliferation and migration of chondrocytes [74,85]. Furthermore, SMSC-140-Exos could successfully prevent cartilage degeneration in an OA rat model [74]. Moreover, Zhang et al. could find that melatonin—a hormone secreted mainly from the pineal gland—prevents OA-induced cartilage degradation via targeting miRNA-140 in an in vitro study [86]. In addition to directly increasing or decreasing the expression of miRNAs, pharmaceutical drugs such as 5,7,3′,4′-Tetramethoxyflavone L(TMF) and Trichostatin A (TSA) can protect chondrocytes by indirectly regulating miRNAs [17,66]. 

A part of the positive effect of miRNA-based therapeutics for OA treatment could be their modulation of the Wnt signaling pathway. Previous studies have revealed the possible application of Wnt pathway-based therapeutics in other pathologies, such as cancer [87], Alzheimer’s disease [88], and even rare diseases like Alkaptonuria [89] but also in OA (Lorecivivint and SM04690 [36,90]). Interestingly, there seems to be a feedback loop of the Wnt-pathway, changing miRNAs expression. For example, the expression of miR-140 in cartilage chondrocytes can be affected by Wnt signaling under hydrostatic pressure [80]. During the development of OA, miR-140 shows dynamic expression and differential effects on cartilage matrix remodeling depending on the stage of OA, which again could be caused by the changing activity of Wnt signaling in the context of OA development [91]. Furthermore, miR-221-3p demonstrated a mechanosensitive character in cartilage as reported by Hecht et al. [92], and several components of the Wnt pathway were identified as direct targets of miR-221-3p in breast cancer [93]. Thus, the ample evidence of interaction of miRNAs, the Wnt pathway, and tissues of the joint promises a new angle to understand the bone–cartilage remodeling in the context of OA. 

Many different miRNA-based treatments have already been proven to be effective at the cellular or animal level, calling for translation of this knowledge to the treatment of patients. Based on current experience in the cancer field, delivery issues, low bioavailability, specific tissue delivery, off-target side effects, miRNA instability, immunogenicity, and even tumorigenicity are major obstacles to be addressed before human clinical trials can be implemented on a large scale [94,95].

## 12. Future Perspective

Over the past decades, the research focus of OA has been shifting from the analysis of single tissues (cartilage and bone) to a more holistic approach of analyzing the interaction of subchondral bone and cartilage. During the development of OA, both compartments demonstrate alterations not only at the tissue level but also at the cellular level. The third important issue in this respect is the synovial tissue, and synovitis induced by destructed cartilage can further accelerate cartilage breakdown and bone remodeling by secreting catabolic and proinflammatory mediators and is thought as a vital cause of clinical symptoms [96]. Wnt signaling and miRNAs participate in the homeostasis of all three tissues and, therefore, in OA progression from at least three different directions. Even in one tissue, Wnt signaling can have dual effects, like in cartilage, where both activation and inhibition of Wnt signaling can induce cartilage destruction in animal models [97]. A potential explanation for the phenomenon might be post-translational regulation by miRNAs during different interventions, which may help to understand the experimental results. Remodeling of cartilage and subchondral bone is tied to mechanics, and the existence of mechanosensitive miRNAs suggests that a dynamic expression of miRNAs and their modulation on Wnt signaling in the joint can be important in this respect. Present studies mostly adopt miRNA mimics or plasmid transfection to increase the expression of specific miRNAs in cells, while further research is necessary to uncover the underlying mechanisms. Accurate and efficient transport of miRNAs to target cells is an important issue to be resolved. Additionally, the transfection method still has limitations when trying to simulate endogenous miRNA expression changes, so there is demand for more sophisticated approaches [98,99]. Moreover, the exact mechanism of how miRNAs are secreted, transferred, uptaken and finally affect the Wnt signaling in recipient cells is unclear and relevant studies are rare. In this context, further exploration of the relationship between miRNAs and Wnt signaling, especially if and how miRNAs can be transported across the interfaces between bone–cartilage and synovial tissue in the context of OA, is required. One possibility for this transport may be through extracellular vesicles (EVs). EVs have been shown to mediate cell–cell communication by transferring cargos containing miRNAs, and the bilayer of EVs can protect miRNAs from RNase-induced degradation during the transportation process. Given the existence of microchannels between cartilage and bone [100,101], EVs mediated miRNAs’ regulation on Wnt signaling could be a center of attraction for innovation and an important area to perform an in-depth study of cell–cell communication between the joint tissues. 

Furthermore, the current surge of genomic and proteomic data generation will increase knowledge in this area. Combining novel delivery platforms like EVs and the increasing knowledge of small RNA clinical trials in other areas (e.g., thalassemia (NCT04718844), hepatitis C (NCT01646489), type 2 diabetes (NCT02612662), multiple solid tumors (NCT01829971), or hyperlipidemias (NCT04606602) will pave the way for these new and innovative therapies. This gives hope that miRNA treatments for OA may become a clinical reality in the near future. 

## 13. Conclusions

The intersection of different regulatory mechanisms can play a vital role in biological processes such as cell proliferation, differentiation, apoptosis, and regeneration. In this review, we summarized the current understanding of the interaction between miRNAs and the canonical Wnt pathway in OA. miRNA-based therapeutics may provide a novel option for OA treatment. A clearer picture of the interaction between miRNAs and the Wnt pathway will deepen our understanding of OA pathogenesis and accelerate the development of these new, hopefully disease-modifying therapeutic targets in the future.

## Figures and Tables

**Figure 1 ijms-22-09887-f001:**
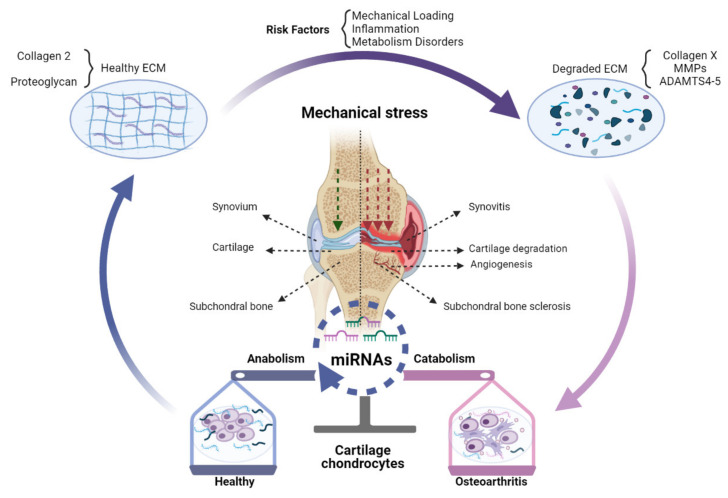
The role of miRNAs in OA pathogenesis. Changed expression of miRNAs in OA chondrocytes can shift the balance of homeostasis, which can impose a protective effect (blue), destructive effect (pink), or both (merged blue and pink) on chondrocytes. A healthy knee joint could synthesize and secrete normal ECM mainly comprised of Collagen 2 and proteoglycan. Risk factors of OA, especially abnormal mechanical stress, inflammation, and metabolic disorders, can destroy ECM and induce excess expression of Collagen X, MMPs, and ADAMTS-4,5, accelerating cartilage destruction. These processes can be epigenetically modulated by miRNAs expressed in cells of the joint tissues, chondrocytes, synoviocytes, and subchondral bone cells. As a result, chondrocyte metabolism can be changed from anabolic towards catabolic, which in turn accelerates the degradation of cartilage together with subchondral bone sclerosis, angiogenesis, and synovitis.

**Figure 2 ijms-22-09887-f002:**
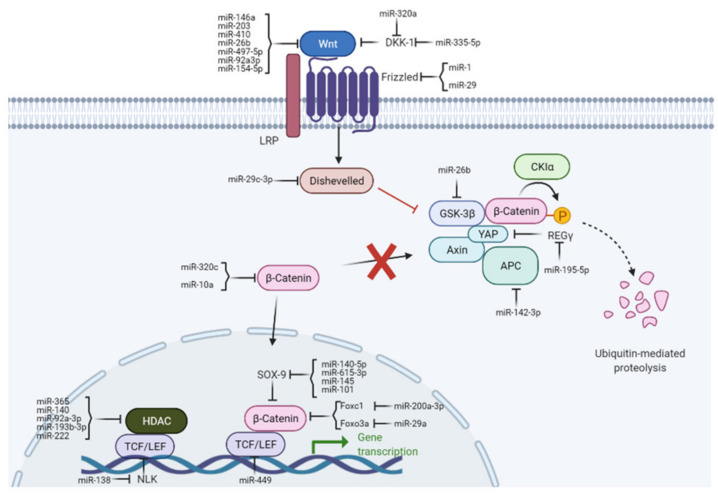
Regulatory functions of miRNAs on canonical Wnt signaling pathways in OA. miRNAs influence various steps of the Wnt pathway by targeting Wnt ligands (i.e., DKK1), receptors (i.e., Wnt), β-catenin, or Wnt transcription factors (i.e., TCF/LEF).

**Table 1 ijms-22-09887-t001:** miRNAs targeting Wnt ligands/receptors and associated proteins in the pathogenesis of OA.

miRNA	Target	Effect of miRNA	Reference
miR-497-5p	Wnt3a	Inhibits cartilage matrix degradation	Hou et al., (2019) [38]
miR-410	Wnt3a	Promotes chondrogenic differentiation	Y. Zhang et al., (2017) [39]
miR-92a-3p	Wnt5a	Inhibits cartilage matrix degradation	Mao et al., (2018) [40]
miR-26b	Wnt	Inhibits chondrogenic differentiation	T. Huang et al., (2019) [41]
miR-154-5p	Wnt11	Inhibits osteogenic differentiation	Li et al., (2015) [42]
miR-203	Mcl-1	Promotes inflammation	Zhao et al., (2017) [43]
miR-146a	CXCR4	Promotes inflammation	Sun et al., (2017) [44]
miR-29	FZD3, FZD5, DVL3, FRAT2, and CK2A2	Complex role in cartilage homeostasis	Le et al., (2016) [45]
miR-1	FZD7	Inhibits cartilage matrix degradation	Xing et al., (2017) [46]
miR-335-5p	DKK1	Promotes osteogenic differentiation	J. Zhang et al., (2011) [47]
miR-320a	BMI-1	Inhibits cartilage matrix degradation	Peng et al., (2017) [48]

**Table 2 ijms-22-09887-t002:** miRNAs targeting β-catenin destruction complex and associated proteins in the pathogenesis of OA.

miRNA	Target	Effect of miRNA	Reference
miR-320c	β-catenin	Inhibits cartilage matrix degradation	Zhang et al., (2019) [53]
miR-10a	β-catenin	Inhibits osteogenic differentiation	Li et al., (2015) [54]
miR-200a-3p	FoxC1	Inhibits cartilage matrix degradation	Wang et al., (2019) [55]
miR-142-3p	APC	Promotes osteogenic differentiation	Hu et al., (2013) [56]
miR-26b	GSK3β	Promotes osteogenic differentiation	Hu et al., (2019) [57]
miR-195-5p	YAP	Promotes chondrocytes apoptosis	Shu et al., (2019) [58]
miR-29c-3p	DVL2	Inhibits osteogenic differentiation	Wang et al., (2018) [59]

**Table 3 ijms-22-09887-t003:** miRNAs targeting Wnt pathway transcription factors and associated proteins in the pathogenesis of OA.

miRNA	Target	Effect of miRNA	Reference
miR-449a	LEF-1	Inhibits chondrogenesis	Paik et al., (2012) [64] Baek et al., (2018) [65]
miR-29a	Foxo3a	Promotes chondrocyte apoptosis	Huang et al., (2019) [66]
miR-365	HDAC-4	Promote chondrogenesis	Guan et al., (2011) [16]
miR-222	HDAC-4	Promote chondrogenesis	Song et al., (2015) [17]
miR-92a-3p	HDAC-2	Promote chondrogenesis	Mao et al., (2017) [67]
miR-193b-3p	HDAC-3	Promote chondrogenesis	Meng et al., (2018) [68]
miR-140	HDAC-4	Promote chondrogenesis	Tuddenham et al., (2006) [15]
miR-138	NLK	Promotes chondrogenesis	Xu et al., (2019) [69]
miR-101	Sox9	Inhibits chondrogenesis	Gao et al., (2019) [70] Dai et al., (2012) [71]
miR-145	Sox9	Inhibits chondrogenesis	Sanchez et al., (2012) [72]
miR-615-3p	Sox9	Inhibits chondrogenesis	Zhou et al., (2018) [73]
miR-140-5p	RaIA	Promotes chondrogenesis	Tao et al., (2017) [74]

**Table 4 ijms-22-09887-t004:** miRNAs regulated by canonical Wnt signaling pathways in the pathogenesis of OA.

Molecules	miRNA	Effect of miRNA	Reference
Wnt/β-catenin	miR-27a/b, miR-140, miR-146a/b miR-365	Promotes cartilage matrix degradation	Cheleschi et al., (2017) [80]
HDAC inhibitor	miR-146a	Inhibits inflammation	Wang et al., (2013) [81]
β-catenin/TCF4/LEF1	miR-29	Promotes chondrocyte apoptosis	Kapinas et al., (2019) [82]
Wnt/β-catenin	miR-140	Promotes chondrogenesis	Yang et al., (2011) [83]

## Data Availability

Not applicable.

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
