# Peer review of "The Interaction between microRNAs and the Wnt/β-Catenin Signaling Pathway in Osteoarthritis"

_ijms, 2021, doi:10.3390/ijms22189887_

Round 1

Reviewer 1 Report

The authors of the review article ‘The Interaction between microRNAs and the Wnt/ß-catenin Signalling Pathway in Osteoarthritis’ have produced a comprehensive and detailed description of the biological role of a large number of miRNAs which interact with the Wnt/ß-catenin signalling pathway and have been implicated in regulating osteoarthritis pathogenesis and progression. I thought this review had synthesised a large amount of information carefully and would be of use to researchers in this field. I found Figure 2 to be particularly helpful in following this article and I found myself regularly referring back to it to see where each miRNA fitted into the pathway. However, this article appeared to largely summarise the literature on each miRNA in turn and did not include much overview of the field, e.g. where the field might go next or the main areas of debate. Without major revisions to include an overarching commentary on the field I am not currently able to recommend this article for publication.

MINOR COMMENTS:

  1. I don’t think that reference 5 on line 46 is correct as a quick search of that article did not find the word ‘osteoarthritis’ or ‘arthritis’ anywhere in the main text.
  2. The gene DCGR8 on line 50 should be called by its full name: DiGeorge Syndrome Critical Region Gene 8, you are missing the word ‘Gene’.
  3. Please explain more fully what you meant by ‘tissue-level prediction’ on line 63, I didn’t really understand what you meant here.
  4. Please state what the promising results are that are described on line 135.
  5. There is a typo on line 182 (‘BesiddeOther’).
  6. The acronym on NSAID on line 325 has not previously been defined. Please check all other acronyms are defined at their first use as well.

Author Response

Reply to Reviewers

We want to thank all the reviewers for their careful review and valuable comments. According to these suggestions we now have thoroughly revised the manuscript as detailed in our point by point answer below.

In the manuscript, revised portions are marked in light yellow.

Reviewer 1:

The authors of the review article ‘The Interaction between microRNAs and the Wnt/ß-catenin Signalling Pathway in Osteoarthritis’ have produced a comprehensive and detailed description of the biological role of a large number of miRNAs which interact with the Wnt/ß-catenin signalling pathway and have been implicated in regulating osteoarthritis pathogenesis and progression. I thought this review had synthesised a large amount of information carefully and would be of use to researchers in this field. I found Figure 2 to be particularly helpful in following this article and I found myself regularly referring back to it to see where each miRNA fitted into the pathway.

Answer: The authors would like to thank the reviewer for this very positive evaluation of our work.

Reviewer 1:

However, this article appeared to largely summarise the literature on each miRNA in turn and did not include much overview of the field, e.g. where the field might go next or the main areas of debate. Without major revisions to include an overarching commentary on the field I am not currently able to recommend this article for publication.

Answer: We would like to thank the reviewer for pointing out this limitation of our work. To remedy this deficiency, we now have included a novel paragraph “12. Future perspective” (Page11-12/Line 420-460), where we give an overview over this field. Additionally, we added a selection of recent clinical trials involving small RNA treatment for several diseases, underlining the importance of small RNA treatment in the near future.

MINOR COMMENTS:

Reviewer 1:

  1. I don’t think that reference 5 on line 46 is correct as a quick search of that article did not find the word ‘osteoarthritis’ or ‘arthritis’ anywhere in the main text.

Answer: We thank the referee very much for pointing our our error. In the revised manuscript, we have replaced this reference with the  appropriate one (Page 2/Line 48, Page 13/Line 494).

Reviewer 1:

  1. The gene DCGR8 on line 50 should be called by its full name: DiGeorge Syndrome Critical Region Gene 8, you are missing the word ‘Gene’.

Answer: Thank you very much for your careful review. We now have corrected this (Page 2/Line 52).

Reviewer 1:

  1. Please explain more fully what you meant by ‘tissue-level prediction’ on line 63, I didn’t really understand what you meant here.

Answer: Thank you for pointing out this unclarity. We now have completely rephrased the paragraph and hope it is easier to understand. (Page 2/Line 63-67).

Reviewer 1:

  1. Please state what the promising results are that are described on line 135.

Answer: Thanks a lot for your kind suggestion. We have added the relevant description of the promising results accordingly (Page 4/Line 175-179).

Reviewer 1:

  1. There is a typo on line 182 (‘BesiddeOther’).

Answer: Thanks. We have corrected this error (Page 6/Line 227).

Reviewer 1:

  1. The acronym on NSAID on line 325 has not previously been defined. Please check all other acronyms are defined at their first use as well.

Answer: Many thanks for your careful review. We now have defined the acronym on NSAID in the revised manuscript (Page 10/Line 379). We also have checked all other acronyms accordingly and confirmed that they were defined before their first use.

Reviewer 2 Report

The paper by Shang et al is an overview of microRNAs involved in the regulation of the Wnt pathway, with a particular attention to the possible therapeutic applications of miRNAs for Osteoarthritis treatment.  The topic is interesting and dealt in detail.  However, this reviewer has some comments.

1. In the Paragraph 2, it is written: ‘More importantly, pathologic expression of miRNAs can lead to various diseases including OA [5].’ The paper cited at the end of this sentence is disjoined, because it is about the miRNA dysregulation only in cancer. We suggest using a more appropriate reference.

2. In the Paragraph 3, authors introduce the topic with the sentence ‘miRNAs mediated epigenetic regulation may play an important part in the transition from healthy joint to OA.’ Why the authors refer only to the epigenetic regulation, considering that the examples reported later don’t involve epigenetic mechanism? Moreover, in the sentence ‘Besides, MMP-19 was identified as the direct target of miR-193b-3p whose expression was decreased in chondrocytes, ...’ is not specified that the miR-193b-3p expression decrease in OA chondrocytes. This clarification could make the concept clearer.

3. In the Paragraph 4, a brief explanation of the Wnt pathway, before analysing its regulation in OA, could enhance the comprehensibility of the topic for non-experts. Moreover, the considerations about the possible treatment of OA through Wnt pathway regulation, could be extended to other pathologies, in which the pathway is still involved, such as cancer (https://pubmed.ncbi.nlm.nih.gov/16683072/), Alzheimer’s disease (https://pubmed.ncbi.nlm.nih.gov/10967351/) or rare diseases as Alkaptonuria (https://pubmed.ncbi.nlm.nih.gov/31989660/).

4. A minor observation: reference 54 in the test, and some references in the Table 3, are between parentheses, instead of the square brackets.

Author Response

Reply to Reviewers

We want to thank all the reviewers for their careful review and valuable comments. According to these suggestions we now have thoroughly revised the manuscript as detailed in our point by point answer below.

In the manuscript, revised portions are marked in light yellow.

Reviewer 2:

The paper by Shang et al is an overview of microRNAs involved in the regulation of the Wnt pathway, with a particular attention to the possible therapeutic applications of miRNAs for Osteoarthritis treatment.  The topic is interesting and dealt in detail. 

Answer: We want to thank the reviewer for the positive evaluation.

Reviewer 2:

However, this reviewer has some comments.

  1. In the Paragraph 2, it is written: ‘More importantly, pathologic expression of miRNAs can lead to various diseases including OA [5].’ The paper cited at the end of this sentence is disjointed because it is about the miRNA dysregulation only in cancer. We suggest using a more appropriate reference.

Answer: We thank the referee for pointing out our error. In the revised manuscript, we have replaced this reference now with the appropriate one (Page 2/Line 48, Page 13/Line 565).

Reviewer 2:

  1. 2. In the Paragraph 3, authors introduce the topic with the sentence ‘miRNAs mediated epigenetic regulation may play an important part in the transition from healthy joint to OA.’ Why the authors refer only to the epigenetic regulation, considering that the examples reported later don’t involve epigenetic mechanism? Moreover, in the sentence ‘Besides, as the MMP-19 was identified direct target of miR-193b-3p whose in expression was decreased chondrocytes, ...’ is not specified that the miR-193b-3p expression decrease in OA chondrocytes. This clarification could make the concept clearer.

Answer: Firstly, we want to thank the reviewer for this important observation. While classical epigenetic mechanisms, such as histone modification and DNA methylation, regulate expression at the transcriptional level, miRNAs function mainly at the posttranscriptional level. miRNAs (as epigenetic modulators) affect protein levels of target mRNAs without modifying their gene sequences. So, the miRNA-mediated regulation of gene expression at the posttranscriptional level belongs to one kind of epigenetic mechanism. After discussion in our group, we decided to modify the text to avoid misunderstanding (Page 2/Line 70). We also modified the sentence ‘Besides, as the MMP-19 was identified direct target of miR-193b-3p whose in expression was decreased chondrocytes, ...’ according to your advice (Page 3/Line 91-93).

Reviewer 2:

  1. In the Paragraph 4, a brief explanation of the Wnt pathway, before analysing its regulation in OA, could enhance the comprehensibility of the topic for non-experts. Moreover, the considerations about the possible treatment of OA through Wnt pathway regulation could be extended to other pathologies, in which the pathway is still involved, such as cancer (https://pubmed.ncbi.nlm.nih.gov/16683072/), Alzheimer’s disease (https://pubmed.ncbi.nlm.nih.gov/10967351/) or rare diseases as Alkaptonuria (https://pubmed.ncbi.nlm.nih.gov/31989660/).

Answer: We thank the referee for the suggestion about the explanation of the Wnt pathway. In the revised manuscript, we have added a paragraph “4. Wnt pathway overview” (Page 3-4/Line 114-146). With respect to the possible treatment of OA through Wnt pathway regulation, we have also added relevant information according to your kind suggestion (Page 11/Line 398-412).

Reviewer 2:

  1. A minor observation: reference 54 in the test, and some references in Table 3, are between parentheses, instead of the square brackets.

Answer: Many thanks for your careful review. We have corrected the mistake and now all references are in square brackets in the revised manuscript .

Reviewer 3 Report

This review aims to briefly describe the interaction between microRNAs and the Wnt/β-catenin signaling pathway in osteoarthritis(OA). Through the intersection of different miRNAs regulatory mechanisms, they can play an important role in cell proliferation, differentiation, apoptosis, and regeneration. It shows that the interaction between miRNA and the canonical Wnt pathway in OA can more clearly understand the role of miRNA in the pathogenesis of OA, and provide miRNA strategies and applications in the treatment of OA.

This review is well organized and easy to read. It is very valuable and provides meaningful insights for readers to understand the interaction of miRNA and Wnt pathway, its role in the molecular pathway of osteoarthritis and the relationship between disease, and the development of miRNA for OA therapy.

The data in the table is clear and easy to understand, and the figure is clearly marked, but what are the risk factors in figure 1? The author can add 2 to 3 more factors in the figure for better understanding.

Author Response

Reply to Reviewers

We want to thank all the reviewers for their careful review and valuable comments. According to these suggestions we now have thoroughly revised the manuscript as detailed in our point by point answer below.

In the manuscript, revised portions are marked in light yellow.

Reviewer 3:

This review aims to briefly describe the interaction between microRNAs and the Wnt/β-catenin signaling pathway in osteoarthritis(OA). Through the intersection of different miRNAs regulatory mechanisms, they can play an important role in cell proliferation, differentiation, apoptosis, and regeneration. It shows that the interaction between miRNA and the canonical Wnt pathway in OA can more clearly understand the role of miRNA in the pathogenesis of OA, and provide miRNA strategies and applications in the treatment of OA.

This review is well organized and easy to read. It is very valuable and provides meaningful insights for readers to understand the interaction of miRNA and Wnt pathway, its role in the molecular pathway of osteoarthritis and the relationship between disease, and the development of miRNA for OA therapy.

The data in the table is clear and easy to understand, and the figure is clearly marked, but what are the risk factors in figure 1? The author can add 2 to 3 more factors in the figure for better understanding.

Answer: We want to thank the reviewer for the very positive evaluation and kind suggestion. We have now added the important risk factors to Figure 1 accordingly (Page 2/Figure 1).

Round 2

Reviewer 1 Report

I thank the authors of this manuscript for their rapid response to my comments. I enjoyed reading your section entitled ‘Future Perspective’ which addressed my major comment from the first round of review. All my minor comments have been addressed to my satisfaction (although the acronym NSAID really should be in your Abbreviation List).

I think this manuscript will be a useful resource for readers wanting to know more about the role of specific miRNAs in osteoarthritis and I would recommend it be accepted for publication in its current form.